# Mean Platelet Volume in Neonatal Sepsis: Meta-Analysis of Observational Studies

**DOI:** 10.3390/children9121821

**Published:** 2022-11-25

**Authors:** Carlos J. Toro-Huamanchumo, Cielo Cabanillas-Ramirez, Carlos Quispe-Vicuña, Jose A. Caballero-Alvarado, Darwin A. León-Figueroa, Nicolás Cruces-Tirado, Joshuan J. Barboza

**Affiliations:** 1Escuela de Medicina, Universidad Cesar Vallejo, Trujillo 13007, Peru; 2Escuela de Medicina, Universidad Peruana de Ciencias Aplicadas, Lima 15023, Peru; 3Unidad de Revisiones Sistemáticas y Meta-Análisis, Tau-Relaped Group, Trujillo 13007, Peru; 4Sociedad Científica San Fernando, Universidad Nacional Mayor de San Marcos, Lima 15081, Peru; 5Facultad de Medicina, Universidad Privada Antenor Orrego, Trujillo 13007, Peru; 6Facultad de Medicina Humana, Universidad de San Martín de Porres, Chiclayo 14000, Peru; 7Facultad de Ciencias de la Salud, Universidad Señor de Sipán, Chiclayo 14006, Peru; 8Vicerrectorado de Investigación, Universidad Norbert Wiener, Lima 15046, Peru

**Keywords:** early onset sepsis, newborn, sepsis, mean platelet volume, infant mortality

## Abstract

Introduction: Early onset neonatal sepsis (EONS), particularly in preterm sepsis, is a potentially fatal issue. Evaluation of mean platelet volume (MPV) as an EONS predictor was the goal. Methods: Four databases were used to conduct a systematic evaluation of cohort and case–control studies. Up till the end of October 2022, 137 articles were found utilizing the search method. Following the review, 12 studies were included. Leukocytes, MPV, platelets, gender, birth weight, gestational age, mortality, and C-reactive protein (CRP) were all taken into account while analyzing the prediction of EONS. Inverse-variance methodology and the random-effects model were used. Using GRADE, the evidence’s quality was evaluated. Results: Neonatal patients with sepsis had significantly higher MPV levels than do neonates without sepsis (MD 1.26; 95% CI 0.89–1.63; *p* < 0.001). An increased MPV during the first 24 h postpartum was associated with high CRP values and high risk of neonatal mortality. In the investigations, the MPV cutoff for sepsis patients was 9.95 (SD 0.843). Overall certainty of the evidence was very low. Conclusions: The increased MPV during the first 24 h postpartum may be predictive of EONS and mortality. Future studies are warranted.

## 1. Introduction

A potentially fatal issue is EONS. It is responsible for up to 30 to 50 percent of all newborn fatalities in underdeveloped nations [1].

Additionally, neonatal sepsis (NS) is regarded as a significant contributor to newborn morbidity and mortality on a global scale [2]. It generates an economic impact, for example, the annual cost of deaths in newborns that are due to sepsis in the United States is 1.97 billion dollars [3]. EONS develops within the newborn’s first 72 h of life [4]. There is a certain degree of difficulty in its diagnosis due to the non-specific clinical manifestations in this type of patient. [5]. This has led to the search for predictors of sepsis. An example is the mean platelet volume (MPV), which could help diagnose EONS [6].

Likewise, some neonatal and maternal risk factors would lead to the development of sepsis in the newborn, such as, low birth weight, chorioamnionitis, prematurity and premature rupture of membranes [7]. The association between neonatal sepsis and other factors, such as gestational age, mode of delivery, sex, and age of the neonate, is still not well defined [8]. The diagnosis of NS is difficult to establish, mainly due to the non-specific clinical features of the patient, where there is temperature instability, tachypnea, hypotension, hypotonia, and abdominal distention [9]. However, some acute phase reactants and inflammatory mediators are predictors of sepsis, such as procalcitonin (PCT), interleukins 6 and 8, C-reactive protein (CRP), and fibrinogen [10]. Due to their significant negative predictive value, platelets have recently been employed as a biomarker for the exclusion of newborn sepsis [11].

According to studies in recent years, among these mediators is the MPV, which could act as a predictor of EONS [12]. The main mechanisms that platelets offer to identify sepsis are the disturbed platelet production during sepsis and the measurement of MPV as a marker of the mean size of circulating platelets in the whole blood count [13,14]. Therefore, the present study aims to evaluate MPV as a predictor of EONS.

## 2. Materials and Methods

### 2.1. Data Sources and Searches

From March to June 2022, we searched PubMed, Embase, Web of Science (WOS), and Scopus. We applied the PRISMA 2020 criteria for reporting this systematic review [15]. The key search terms used were “mean platelet volume” and “neonatal sepsis”. Abstracts from cohort and case–control studies that assessed MPV as a predictor of EONS were included. There was no restriction based on the year of publication.

Reviews, case reports and case series studies, editorials, meta-analysis, and letters were excluded.

### 2.2. Study Selection

In this study, we employed the P-O criteria, considering the Population (Newborns with an early onset neonatal sepsis diagnosis) and the Outcome (MPV measured by mean and standard deviation). Cohort and case–control studies that evaluated mean platelet volume as a predictor of EONS were included in the study. Case-control studies that matched the first two criteria were also included. We omitted narrative and systematic reviews, experiments on animals, clinical trials, case reports, abstracts from conferences, and letters.

Using Rayyan QCRI (https://rayyan.qcri.org/ (accessed on 20 June 2022)), two writers (C.J.T.H. and C.C.R.) independently reviewed the titles and abstracts in accordance with the inclusion and exclusion criteria [16]. The full texts of chosen pertinent studies were searched for in-depth examination. Conflicts were settled by consensus and, if necessary, input from a third author (J.J.B.). Selected articles were stored using the Endnote 20 program.

### 2.3. Outcomes

The outcome was the Mean platelet volume (fL), measured in mean and standard deviation (SD).

### 2.4. Data Extraction

Two authors (NCT and JJB) separately extracted the information using the prescribed forms. Consensus was used to resolve disagreements, and a third author was consulted if necessary. The initial author’s name, the year, the study’s design, the participants’ number, the target MPV (fL), the CRP, the mortality, and the study’s results were all gathered. Conflicts were settled with the assistance of a third author.

### 2.5. Risk of Bias (RoB) Assessment

The RoB was then assessed by two investigators. Finally, the Newcastle Ottawa Scale (NOS) tool was used to examine both cohort and case–control studies independently. This scale was created to evaluate the quality of nonrandomized trials to incorporate quality judgments into meta-analysis interpretation. In the interpretation of meta-analyses, the NOS evaluated quality based on content, design, and simplicity of use. It is made up of eight pieces that are separated into three dimensions (comparison, selection, type of study).

### 2.6. Statistical Analysis

For meta-analysis, random-effects models and inverse-variance method were applied. For continuous outcomes, the effects of mean platelet volume (MPV) and other characteristics were characterized using mean differences (MD) with 95 percent confidence intervals (95% CI). The I^2^ statistic was used to look at study heterogeneity: 0–30 percent means low, 30–60 percent means moderate, and >60 percent means high. The primary outcomes were broken down into subgroups based on gestational age (preterm versus term newborns). The R 3.5.1 meta library’s metabin and metacont functions were utilized.

### 2.7. Quality of Evidence

GRADE was evaluated for the certainty of evidence [17]. A summary results table was created utilizing the results using the GRADEpro software (McMaster University and Evidence Prime Inc., Hamilton, ON, Canada).

## 3. Results

### 3.1. Selection of Studies

In first place, a total of 137 records were found in the four databases of which 69 duplicates were eliminated. Second, 68 records were selected by title and abstract, and then just 29 records were evaluated in full text. The 37 excluded did not follow our inclusion or exclusion criteria, or were case reports, letters to editor, narrative review or editorial. Finally, 12 studies were ultimately included in the systematic review [18,19,20,21,22,23,24,25,26,27,28,29]. (Figure 1).

### 3.2. Characteristics of Included Studies

Included studies (Table 1) according to the design were nine case–control studies and only one cohort study conducted in countries such as Egypt, Turkey, Iran, and India. The most common concept regarding the definition of sepsis among all studies was: “Proven sepsis” [18,19,20,21,22,23,24,25,26,27], as clinical signs of sepsis with isolation of pathogen in blood, cerebrospinal fluid (CSF), or urine; and, “Probable sepsis” [18,19,20,21,22,23,24,25,26,27], as clinical signs of sepsis without isolated pathogen, with one or more of the following criteria: (a) maternal fever, foul-smelling fluid, prolonged rupture of membranes more than 12 h, gastric polymorph count of more than 6/hpf; (b) positive sepsis screen with 2 of the following 4 parameters: total leukocyte count <5000/mm^3^, the ratio of band cells to total neutrophils equal to or greater than 0.2, CRP greater than or equal to 0.6 mg/dl, micro Erythrocyte Sedimentation Rate (ESR) greater than or equal to 15 mm at the end of the first hour; and (c) radiological evidence of pneumonia. Most studies agrees that high MPV is predictive of early neonatal sepsis and can function as an adequate biomarker.

### 3.3. Risk of Bias Analysis

After the evaluation with the Newcastle Ottawa Scale tool, four studies [19,21,23,25] showed a high risk of bias, and six showed a low level of risk of bias according to the assessment.

### 3.4. Effects of Mean Platelet Volume in Sepsis

We identified that the MPV is significantly higher in neonates with sepsis (MD 1.26; 95% CI 0.89–1.63; *p* < 0.001; Figure 2), compared with neonates without sepsis.

### 3.5. Cut-Off and Diagnosis Accuracy between Studies

The cut-off of MPV in patients with sepsis among the studies was 9.95 (SD 0.84). Only four studies had a sensitivity greater than 80% [18,22,25,26]. Similarly, six studies had a specificity greater than 80% [18,21,24,25,27,29]. The Area under curve (AUC) of most studies was greater than 0.6 [18,20,21,22,23,24,25], and two studies [27,28], was less than 0.5 (Table 2).

### 3.6. Quality of Evidence

We assess the quality of evidence using the GRADE tool (Table 3). Overall certainty of the evidence was very low. The risk of bias decreases by two levels because there are more than two studies with a high risk of bias. Additionally, the inconsistency was decreased by two levels because heterogeneity is higher than 60%. The indirectness and impression were not serious.

## 4. Discussion

Our study found an increase in MPV in the first 24 h in neonates with sepsis compared to without sepsis group. Therefore, elevated MPV values are predictive of EONS. In addition, we noted an increase of mortality and increased CRP values in patients with neonatal sepsis.

Blood culture is the gold standard for diagnosing newborn sepsis, which has some inefficiencies, such as a high false positive rate (due to contamination) and possible difficulty in obtaining results within 48 to 72 h [30]. Due to the deficiencies above, it was decided to look for alternatives. Here, the possibility of using inflammatory mediators appears, as in the case of MPV, which allows the diagnosis, follow-up, and prediction of the severity of sepsis in neonates [31].

In the pathophysiology of sepsis, there will be an alteration in the coagulation cascade, which will allow the release of multiple pro-and anti-inflammatory cytokines, leading to thrombus formation [32]. This will cause fibrinolytic and fibrinogenic substances to be depleted, leading to further platelet destruction. Finally, the bone marrow will increase the production of young platelets, which are larger and functionally more active, expressing themselves in an increase in vascular endothelial growth factor (VEGF) [33]. The bone marrow will then express itself in an increase in MPV, thus predicting the onset of sepsis.

Platelets have a significant role in sepsis-induced coagulopathy in septic newborns. P-selectin, which is expressed on the surface of platelets during systemic inflammation, enhances platelet adherence to leukocytes and platelet aggregation in addition to tissue factor expression on monocytes [34]. In septic infants, platelet consumption and thrombus development through active endothelium cause thrombocytopenia [35]. C-reactive protein reaches its peak after fifty hours and helps complement bind to foreign or damaged cells in response to inflammation. In addition to clinical evidence, CRP, a non-specific illness response, can aid in the diagnosis of septicemia [36].

It should be mentioned that no previous systematic reviews were found with which this work can be compared. On the other hand, we searched four databases, after which we included ten studies, and, in addition, we assessed the risk of bias using the NOS tool. Regarding the statistical analysis, our study analyzed the continuous variables with the mean difference and standard deviation according to the homogeneity of the measurement factor. Additionally, the random-effects model was applied.

The Importance of our findings lies in the fact that MPV is routinely reported as part of the complete blood count. However, a midpoint for EONS prediction could not be found because the studies reported MPV values with different measurement techniques.

Most of the studies evaluated reported values above the established cut-off, which may be associated with the high sensitivity and specificity reported among the studies. With the values established as an average, it is possible to have an estimated cut-off for evaluating patients with a potential diagnosis of neonatal sepsis [37]. Other studies also reported cut-off values in other population settings in sepsis [38,39,40].

Likewise, the low cost and easy availability makes the MPV a marker with greater significance. Another critical point to highlight is the homogeneity of the studies, nine of which were of the case–control type, allowing greater homogeneity in the groups studied. Associated factors, such as prematurity or low birth weight, were considered in very few studies, so their true significance could not be determined.

The study has some limitations. First, observational studies are subject to confounding variables, thus increasing statistical heterogeneity between studies, and may influence the true overall effect. Second, there is a very low certainty of the evidence, which limits an adequate interpretation despite the significant effect. However, this certainty of the evidence is not related to interventions but as a predictor.

## 5. Conclusions

In conclusion, with the results of this study, we can affirm that increased MPV may be used as a predictor of EONS, which would potentially allow us to avoid excessive antibiotic use and unnecessary hospitalizations.

## Figures and Tables

**Figure 1 children-09-01821-f001:**
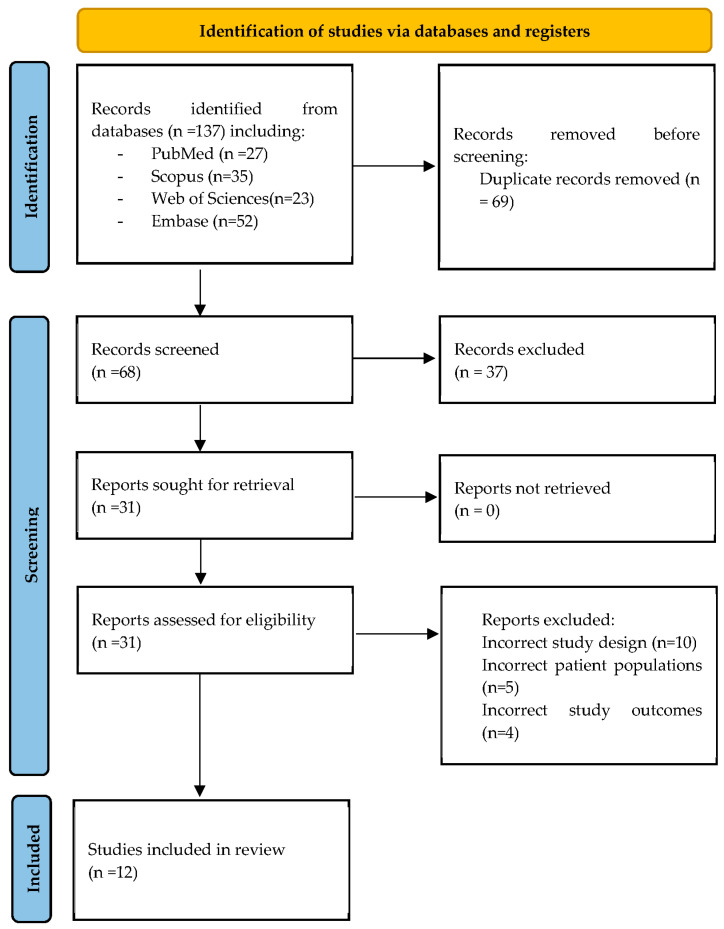
PRISMA flow chart of the studies selection process.

**Figure 2 children-09-01821-f002:**
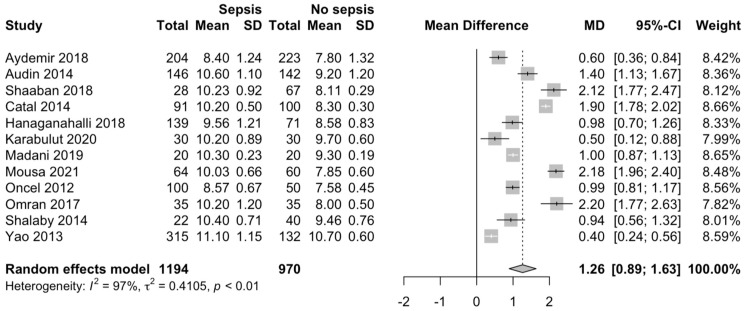
MPV in neonates with and without EONS [18,19,20,21,22,23,24,25,26,27,28,29].

**Table 1 children-09-01821-t001:** Characteristics of Included studies.

Author, Year	Country	Study Design	Definition of Sepsis	MPV (Mean, SD)	CRP (Mean, SD)	Mortality (n, %)	Conclusions	Risk of Bias
Shaaban, 2018 [18]	Egypt	Cohort study	Clinical diagnosis	10.23 ± 0.92	14.7 (12 - 24)	10(35.7%)	The MPV has a high sensitivity and specificity for establishing the diagnosis of snit. In addition, it can be used as a predictor of mortality in preterm infants.	Low
Oncel, 2012 [19]	Turkey	Case-control study	Clinical and laboratory diagnosis	8.57 ± 0.67	27.4 (0.11–45.2)	NR	The MPV values are significantly higher in patients with sepsis, proven or clinical, compared to healthy controls.	High
Omran, 2017 [20]	Egypt	Case-control study	Clinical and laboratory diagnosis	10.2 ± 1.2 fl	27.3 ± 13.3 mg/L	NR	The MPV, taking 10.2 fl as a cut-off point, is able to predict neonatal sepsis with a sensitivity and specificity of 80%.	Low
Mousa, 2021 [21]	Egypt	Case-control study	Clinical and laboratory diagnosis	10.03 ± 0.66	57.47 ± 56.9	NR	The MPV is a cost-effective platelet parameter for predicting the diagnosis of neonatal sepsis.	High
Karabulut, 2020 [22]	Turkey	Case-control study	Clinical and laboratory diagnosis	10.2 ± 0.89	17.3 ± 6.6	NR	The MPV is a biomarker, which at the cut-off value of 9.3 fl, has a sensitivity of 84% and a specificity of 32% for the diagnosis of EONS.	Low
Madani, 2019 [23]	Iran	Case-control study	Clinical diagnosis	10.3 ± 0.23	NR	NR	Newborns with sepsis have significantly higher MPV values, with a sensitivity of 65.3% and a specificity of 75% for diagnosing this pathology.	High
Hanaganahalli 2018 [24]	India	Case-control study	Clinical diagnosis	Group i = 9.56 ± 1.21 group ii = 8.86 ± 0.98	Group i = 66.95 ± 74.78 group ii = 18.01 ± 3.39	NR	The MPV, with a cutoff value ≥ 9.5 fl, can predict the diagnosis of neonatal sepsis with a sensitivity of 85% and a specificity of 56%.	Low
Catal, 2014 [25]	Turkey	Case-control study	Clinical diagnosis	10.2 ± 0.5	5.7 (16.5)	NR	MPV is effective for the diagnosis and follow-up of neonatal sepsis in preterm infants. However, it would not be useful for discriminating sepsis according to time of onset (early or late onset).	High
Aydin, 2014 [26]	Turkey	Case-control study	Clinical and laboratory diagnosis	Group i = 10.6 ± 1.1 group ii = 10.4 ± 0.9	Group i = 33.0 ± 3.4 group ii = 54.6 ± 5.4	NR	High MPV values were not associated with the development of sepsis, as this could be explained by the increase of young platelets in the circulation due to the destruction of platelets present in this pathology.	Low
Aydemir, 2018 [27]	Turkey	Case-control studies	Clinical and laboratory diagnosis	8.4 (5.9–12.1)	15.0 (0.0–200.0)	42.0 (20.6)	MPV values were found to be significantly higher in preterm infants with sepsis. Furthermore, these values were higher in preterm infants with late-onset neonatal sepsis than in those with EONS.	Low
Shalaby, 2013 [28]	Egypt	Case-control study	Clinical diagnosis	10.4 ± 0.71	44.5 ± 12.6	NA	MPV could be assessed in the early diagnosis of neonatal sepsis while SUA level has lower sensitivity in neonatal sepsis.	Low
Yao, 2014 [29]	China	Retrospective cohort	Clinical and laboratory diagnosis	11.1 ± 1.15	5.1 ± 2.7	NA	The diagnostic accuracy of CRP for neonatal sepsis is superior to those of the percentage of neutrophils and MPV	Low

MPV: Mean platelet volume. CRP: C-Reactive Protein. PCT: Procalcitonin. CSF: Cerebrospinal fluid. ESR: Erythrocyte Sedimentation Rate. EONS: Early onset neonatal sepsis. NR: Not Reported.

**Table 2 children-09-01821-t002:** Diagnosis accuracy between studies.

Author, Year	Cut-Off (fL)	Sensitivity (%)	Specificity (%)	NPV (%)	PPV (%)	AUC
Shaaban, 2018 [18]	8.6	97.14	100	NR	NR	0.971
Oncel, 2012 [19]	NR	NR	NR	NR	NR	NR
Omran, 2017 [20]	10.2	80	80	NR	NR	0.873
Mousa, 2021 [21]	10.5	61.1	88.1	NR	NR	0.78
Karabulut, 2020 [22]	9.3	84	32	66.6	55.2	0.666
Madani, 2019 [23]	9.95	65.3	75	27.78	31.81	0.73
Hanaganahalli 2018 [24]	9.5	56	85	88.5	48.3	0.734
Catal, 2014 [25]	10.75	95.2	84	NR	NR	0.944
Aydin, 2014 [26]	NR	82	54	63	76	NR
Aydemir, 2018 [27]	9.1	30.2	88.3	72.7	55.2	0.034
Shalaby, 2014 [28]	10.2	71	63	59	74	0.68
Yao, 2014 [29]	11.4	40.5	88.4	NR	NR	NR

NPV: Negative predictive value. PPV: Positive predictive value. AUC: Area under the curve. NR: Not Reported.

**Table 3 children-09-01821-t003:** GRADE summary of findings table.

Mean Platelet Volume Compared to Control in Early Onset Sepsis Detection Bibliography:											
Certainty Assessment							Summary of Findings				
Participants (Studies) Follow-Up	Risk of Bias	Inconsistency	Indirectness	Imprecision	Publication Bias	Overall Certainty of Evidence	Study Event Rates (%)		Relative Effect (95% CI)	Anticipated Absolute Effects	
							With Control	With Mean Platelet Volume		Risk with Control	Risk Difference with Mean Platelet Volume
Mean Platelet Volume (Assessed with: fl)											
1655 (10 observational studies)	very serious ^a^	very serious ^b^	not serious	not serious	none	⨁◯◯◯ Very low	798	857	-	The mean platelet volume was 0 fl	MD 1.38 fl higher (1.01 higher to 1.76 higher)

CI: confidence interval; MD: mean difference. Explanations: ^a^. ⨁ Decreases two levels because there are more than two studies with high risk of bias. ^b^. ◯ Decreases two levels because heterogeneity is higher than 60%.

## Data Availability

Not applicable.

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
