# Peer review of "Mean Platelet Volume in Neonatal Sepsis: Meta-Analysis of Observational Studies"

_children, 2022, doi:10.3390/children9121821_

Round 1

Reviewer 1 Report

Dear Author, 

You work Mean platelet volume as a predictor of early-onset neonatal sepsis: a systematic review and meta-analysis is great. 

My decision is accept

Author Response

Thanks for your comments

Reviewer 2 Report

In the present study, authors have determined the mean platelet volume as an early marker of neonatal sepsis. My comments are as follows:

1)      Introduction, results and discussion should be adequately described.

2)      All the tables require formatting

3)      All the figures are blurry 

Author Response

Dear reviewer, 

We updated the study according to your comments

Reviewer 3 Report

Toro-Huamanchumo et al submitted a systematic review assessing mean platelet volume as a biomarker predictor of early-onset neonatal sepsis, using only 10 published studies from several databases. Overall, the manuscript is interesting but needs an extensive editing of English and to be more organized as it becomes difficult to follow the main concept of the study; the number of studies included was insufficient for a systematic review article. Here, I would suggest some points that might improve the manuscript.

Abstract:

1-      Too much information in the abstract and I would suggest including only 200 words. Please merge the aim of the study with the introduction.

2-      Please write the full name for EONS (neonatal sepsis).

3-      This sentence needs to be grammatically fixed “The increased MPV during the first 24 hours is predictive of EONS and was associated with increased CRP values and increased risk of neonatal mortality”.

4-      It would be great if authors include a paragraph discussing the importance of the study.

Introduction:

1-      Third sentence is a repetition of the first sentence. Please fix it.

2-      I would suggest deleting this sentence “which has led to the development of newborn sepsis in the first 72 hours of life” as you have mentioned it in the previous sentence.

Methods:

1-      It would be better to share with the readers the specified forms that has been used for data extraction.

2-      What did you mean by 165510 observational studies in the grade quality of evidence?

3-      I believe that Table 4 should be 2, and I would suggest using a small font to get words together.

4-      The word “mean” has been duplicated in table 4 column 11. Please fix it.

Results:

1-      Based on what you have excluded 39 out of 68 studies? Were these studies case reports, letters to editor, narrative review, or editorial?

2-      In fig 1., there is a repetition of the phrase “reports excluded”. Please fix it.

3-      Please write the full name of LCR and vsg as first mentioned int the text.

4-      Please rewrite this sentence to be more scientific “Most studies agree that high MPV is predictive of early neonatal sepsis and that it is an adequate biomarker; nevertheless, a single study concludes that high MPV values were not significant”.

5-      Please check table 2 “Characteristics of Included studies” as it has some typos mistakes. For example, MPV media; il-6; snit??

6-      Fig 2 looks like a table, so it’d be better to be a table.

7-      In fig 3, I would suggest adding MPV instead of just a mean and changing the title to be “The mean platelet volume in septic and non-septic neonates”.

8-      Pleas delete the author names and keep only the reference numbers. (Only four studies had a sensitivity greater than 80% (Shaaban et al, Karabulut et al, Catal et al, and Aydin et al.) [18,22,25,26]). (Similarly, five studies had a specificity greater than 80% (Shaaban et al, Mousa et al, Hanaganahalli et al, Catal et al, Aydemir et al.) [18,21,24,25,27]).

9-      Please write the full name of AUC as first mentioned int the text.

Discussion and conclusion:

1.       It is great to recall your results in the discussion, but it would much better in addition to this that you provide a scientific explanation why CRP values are usually increased in neonates with sepsis.

2.       Which coagulation cascade is altered in the septic neonates and what is the exact alteration? Are there certain clotting factors become defect/deficient?

3.       Please rephrase this sentence to be more scientific “Likewise, thanks to its low cost and easy availability, it has become a marker with greater significance”.

4.       A limitation paragraph should be included either at the end of the discussion or in the conclusion. A one caveat of this study is the number of the study, etc.

I wish the authors the best of luck with their revision process.

Author Response

Thank you for your valuable suggestions and remarks. After incorporating your feedback we consider that our manuscript has greatly improved. Please find below the point-by-point responses to your comments. We hope that the present version of the manuscript will be acceptable for publication, and we look forward to your feedback.

Abstract

  1. Too much information in the abstract and I would suggest including only 200 words. Please merge the aim of the study with the introduction.

Response:

Thank you for the observation. We reduced it into 196 words.

“Introduction: Early onset sepsis (EONS) is potentially life-threatening problem especially in preterm. The aim of this systematic review is to evaluate if the mean platelet volume can be a predictor of early-onset neonatal sepsis. Objective: To evaluate MPV as a predictor of EONS. Methods: A systematic review of cohort and case-control studies was performed using PubMed, Scopus, Web of Science, and Embase databases. A total of 137 articles were retrieved using the search strategy up to the end of June 2022. After the evaluation, 10 studies were included. Prediction of EONS was analyzed; the outcomes considered were gestational age, gender, birth weight, mortality, leukocytes, platelets, MPV, and C-reactive protein (CRP). Random-effects model and inverse-variance method were applied. Newcastle Ottawa Scale was performed for risk of bias assessment. Quality of evidence was assessed using GRADE. Results: MPV is significantly higher in neonates with sepsis compared to non-septic neonates (MD 1.38; 95%CI 1.01-1.76; p<0.0001; I2= 96.7%). The cut-off of MPV in patients with sepsis among the studies was 9.7 (SD 0.73). Conclusions: The increased MPV during the first 24 hours is predictive of EONS. Also, it was associated with high CRP values and high risk of neonatal mortality.”

  1. Please write the full name for EONS (neonatal sepsis).

Response:

Thank you for the observation. We corrected it.

  1. This sentence needs to be grammatically fixed “The increased MPV during the first 24 hours is predictive of EONS and was associated with increased CRP values and increased risk of neonatal mortality”.

Response:

Thank you for the observation. We corrected it.

“The increased MPV during the first 24 hours is predictive of EONS. Also, it and was associated with high CRP values and high increased risk of neonatal mortality.”

  1. It would be great if authors include a paragraph discussing the importance of the study.

Response:

Thank you for the observation. The limited number of words for the abstracts allow us to put it in the first part:

“Early onset sepsis (EONS) is potentially life-threatening problem especially in preterm. The aim of this systematic review is to evaluate if the mean platelet volume can be a predictor of early-onset neonatal sepsis.”

Introduction

  • Third sentence is a repetition of the first sentence. Please fix it

Response:

Thank you for the observation. We corrected it.

  • I would suggest deleting this sentence “which has led to the development of newborn sepsis in the first 72 hours of life” as you have mentioned it in the previous sentence.

Response:

Thank you for the observation. We corrected it.

Methods

  • It would be better to share with the readers the specified forms that has been used for data extraction.

Response:

Thank you for the observation. We include the table of extraction that we used to perform the study. We attached the excel archive in the present communication.

  • What did you mean by 165510 observational studies in the grade quality of evidence?

Response:

Thank you for the observation. The table says “1655 (10 observational studies)”.

  • I believe that Table 4 should be 2, and I would suggest using a small font to get words together.

Response:

Thank you for the observation. We corrected it.

  • The word “mean” has been duplicated in table 4 column 11. Please fix it.

Response:

Thank you for the observation. We corrected it.

Results

  1. Based on what you have excluded 39 out of 68 studies? Were these studies case reports, letters to editor, narrative review, or editorial?

Response:

Thank you for the observation. We included this information: “The 39 excluded did not follow our inclusion or exclusion criteria, or were case reports, letters to editor, narrative review or editorial.

  1. In fig 1., there is a repetition of the phrase “reports excluded”. Please fix it.

Response:

Thank you for the observation. We corrected it.

  1. Please write the full name of LCR and vsg as first mentioned int the text.

Response:

Thank you for the observation. We corrected and now is properly mentioned in the English abbreviation.

“Cerebrospinal fluid (CSF)” and “Erythrocyte Sedimentation Rate (ESR)”

  1. Please rewrite this sentence to be more scientific “Most studies agree that high MPV is predictive of early neonatal sepsis and that it is an adequate biomarker; nevertheless, a single study concludes that high MPV values were not significant”.

Response:

Thank you for the observation. We rewrite it:

The majority of studies agrees that high MPV is predictive of early neonatal sepsis and can function as an adequate biomarker. However, one study concludes that high MPV values were not significant.”

  1. Please check table 2 “Characteristics of Included studies” as it has some typos mistakes. For example, MPV media; il-6; snit??

Response:

Thank you for the observation. We corrected it.

  1. Fig 2 looks like a table, so it’d be better to be a table.

Response:

Thank you for the observation. We corrected it.

  1. In fig 3, I would suggest adding MPV instead of just a mean and changing the title to be “The mean platelet volume in septic and non-septic neonates”.

Response:

Dear reviewer, thank you for your suggestion, however, the term "mean difference" in the forest plot is part of the figure and cannot be changed, since this concept is related to the measure of effect used for this outcome or variable, so it is not feasible to switch to "MPV". However, we changed the title.

  1. Pleas delete the author names and keep only the reference numbers. (Only four studies had a sensitivity greater than 80% (Shaaban et al, Karabulut et al, Catal et al, and Aydin et al.) [18,22,25,26]). (Similarly, five studies had a specificity greater than 80% (Shaaban et al, Mousa et al, Hanaganahalli et al, Catal et al, Aydemir et al.) [18,21,24,25,27]).

Response:

Thank you for the observation. We corrected it.

  1. Please write the full name of AUC as first mentioned int the text.

Response:

Thank you for the observation. We corrected it: “Area under curve (AUC)”

Discussion and conclusion

  • It is great to recall your results in the discussion, but it would much better in addition to this that you provide a scientific explanation why CRP values are usually increased in neonates with sepsis.

Response:

Thank you for the observation. We included a paragraph explaining about this.

“C-reactive protein helps in complement binding to foreign or damaged cells in response to inflammation and rising to peak levels after fifty hours. CRP is a non-specific response to disease and, along with clinical evidence, can help in the diagnosis of septicaemia.” (https://pubmed.ncbi.nlm.nih.gov/20110818/ )

  • Which coagulation cascade is altered in the septic neonates and what is the exact alteration? Are there certain clotting factors become defect/deficient?

Response:

Thank you for the observation. We included a paragraph explaining about this.

“In the case of septic neonates, platelets are major players in sepsis-induced coagulopathy. P-selectin is expressed on the platelet surface during systemic inflammation, and it facilitates the platelets’ adhesion to leukocytes and platelet aggregation, in parallel with tissue factor expression on monocytes (https://pubmed.ncbi.nlm.nih.gov/28033029/). Thrombocytopenia in septic newborns is generated by platelet consumption and thrombus formation through the activated endothelium (https://pubmed.ncbi.nlm.nih.gov/29193703/ ).”

  • Please rephrase this sentence to be more scientific “Likewise, thanks to its low cost and easy availability, it has become a marker with greater significance”.

Response:

Thank you for the observation. We corrected it.

“Likewise, the low cost and easy availability, makes the MPV a marker with greater significance.”

  • A limitation paragraph should be included either at the end of the discussion or in the conclusion. A one caveat of this study is the number of the study, etc.

Response:

Thank you for the observation. We included a paragraph at the end of the discussion.

“The study has some limitations. First, observational studies are subject to confounding variables, thus increasing statistical heterogeneity between studies, and may influence the true overall effect. Second, there is a very low certainty of the evidence, which limits an adequate interpretation despite the significant effect. However, this certainty of the evidence is not related to interventions but as a predictor.”

Reviewer 4 Report

Authors reported a systematic review to evaluate if the mean platelet volume can be a predictor of early-onset neonatal sepsis.

This manuscript is potentially interesting, several issues arise.

Abstract: What are EONS and MD?

Table 1 should be written clearly.

The order of Table was not correct.

Table 4 is not clear.

Table 2 should be written clearly.

Standardization of MPV should be discussed.

Effect of blood transfusion should be discussed.

Author Response

Dear reviewer, 

We updated our study according to your comments.

Reviewer 5 Report

Review for the article “Mean platelet volume as a predictor of early-onset neonatal sepsis: a systematic review and meta-analysis”

The paper entitled “Mean platelet volume as a predictor of early-onset neonatal sepsis: a systematic review and meta-analysis” want to evaluate that mean platelet volume (MPV) could be a predictor of early onset sepsis.

The authors used different scientific data bases, PubMed, Scopus, Web of Science, and Embase databases. They found that MPV is significantly higher in neonates with sepsis compared to non-septic neonates. Also, tha increased MPV during the first 24 hours was associated with increased CRP values and increased risk of neonatal mortality.

The subject is very interesting because sepsis represents one of the most common complications with higher mortality in intensive care units (ICUs). So, it is very important to identify possible risk factors in order to improve the prevention methods. The results obtained in the study are very promising for clinicians.The presentation is clear. The writing style is clear.

Author Response

Thanks

Round 2

Reviewer 2 Report

1)      Authors have not provided the revised manuscript with highlighted changes and point-wise replies to the concerns raised.

2)      Tables still require formatting (Even a single word in the sub-headings is coming in two lines, e.g.; Country, participants)

3)      Figure1 is still blurred. Why the line in the second block (Record removed before screening) is half italicized??

Author Response

Dear reviewer:

1)      Authors have not provided the revised manuscript with highlighted changes and point-wise replies to the concerns raised.

We applied all comments and updated the manuscript with the editor's suggestions. I enclose the updated manuscript. 

2)      Tables still require formatting (Even a single word in the sub-headings is coming in two lines, e.g.; Country, participants)

Thanks, we updated the tables and remove duplicated information.

3)      Figure1 is still blurred. Why the line in the second block (Record removed before screening) is half italicized??

Thanks, but is the PDF version. We upload the document in word, and Figure 1 is correct. 

Reviewer 3 Report

The manuscript has been much improved over the first round and all comments have been properly addressed. 

Author Response

Thanks

Reviewer 4 Report

Revised manuscript has been improved, however, several issues retained.

Table 2 can be improved to compact size.

Definition and criteria might be moved to legends.

Table 4 can be improved.

Author Response

Dear reviewer

Table 2 can be improved to a compact size.

A: Thanks. We updated table 2.

Definition and criteria might be moved to legends.

Thanks, we applied the comment.

Table 4 can be improved.

Thanks, we updated table 4.

Round 3

Reviewer 2 Report

1) Tables are not clearly formatted and still the single word is coming in the multiple lines. If you are not able to do it due to space scarcity, instead of Portrait mode you can make tables in the landscape.

2) Authors need to ensure that Figure 1 and other figures are not blurry

Author Response

Dear Reviewer, 2.

  1. Reviewer says: “1) Tables are not clearly formatted and still the single word is coming in the multiple lines. If you are not able to do it due to space scarcity, instead of Portrait mode you can make tables in the landscape”

Our response: " Thank you very much for your recommendations, each table has been organized according to the format of the magazine”.

  1. Reviewer says: Authors need to ensure that Figure 1 and other figures are not blurry.

Our response: " Thank you very much for your recommendations, the quality of the figures has been modified and corrected so that the content presented can be seen.”
